# Maternal Outcomes Following Active vs. Expectant Management of Previable Preterm Pre-Labor Rupture of Membranes: A Meta-Analysis

**DOI:** 10.3390/children10081347

**Published:** 2023-08-04

**Authors:** Megan A. Sylvester, Gabrielle Mintz, Giovanni Sisti

**Affiliations:** Division of Maternal-Fetal Medicine, Department of Obstetrics and Gynecology, College of Medicine-Tucson, University of Arizona, 1501 N. Campbell Avenue, Tucson, AZ 85724, USA; ahernm@arizona.edu (M.A.S.); gamintz@arizona.edu (G.M.)

**Keywords:** active management, chorioamnionitis, elective termination, expectant management, maternal risk, postpartum hemorrhage, preterm PROM, pregnancy, previable, preterm, sepsis

## Abstract

The diagnosis of previable preterm pre-labor rupture of membranes (PROM) is known to be associated with poor outcomes for both the mother and the fetus. Following previable preterm PROM, patients are generally offered either active management through the termination of the pregnancy or expectant management to increase the chances of fetal survival. It is difficult to counsel patients because there is a lack of data directly comparing maternal outcomes following active vs. expectant management. Using the data in the current literature, the goal of the present meta-analysis was to determine if there were any differences in terms of maternal risks when active versus elective management was chosen. PubMed, Google Scholar, EMBASE, and Scopus were searched. We found four studies accounting for a total of 506 patients. The risk ratio (RR) of chorioamnionitis in active vs. expectant management was 0.30 (with a 95% confidence interval, CI, of 0.09–1.02). The heterogeneity of the study results was 81% (I^2^). A sub–analysis of two included studies revealed an RR of postpartum hemorrhage in active vs. expectant management of 0.75 (95% CI 0.27–2.07) and an RR of maternal sepsis of 0.23 (95% CI 0.04–1.28). The heterogeneity of the study results for this sub-analysis was 68% (I^2^) for postpartum hemorrhage and 0% (I^2^) for maternal sepsis. Overall, there was no statistically significant difference in the risk of chorioamnionitis, postpartum hemorrhage, or maternal sepsis when active management was chosen over expectant management in previable preterm PROM at <24 weeks. The scarcity and the high heterogeneity of the available data likely contributed to the lack of statistical significance and calls for further work directly comparing maternal outcomes following active vs. expectant management.

## 1. Introduction

Premature rupture of membranes (PROM) is the rupture of the amniotic membrane before the onset of labor. When rupture occurs before 37 weeks of gestation, it is referred to as preterm PROM [1].

The causes of PROM and preterm PROM sometimes cannot be clearly identified and are therefore called “idiopathic”. When an etiology is identified, it most often falls among the categories of intrinsic or extrinsic factors. Intrinsic factors are, for example, a history of previous PROM or preterm PROM, anatomical uterine malformation, or a constitutionally weak cervical tissue, while extrinsic factors are largely infection or trauma [1].

At term, PROM complicates approximately 8% of pregnancies, whereas preterm PROM complicates roughly 1–3% of pregnancies in the United States (U.S.). Less than 10% of preterm PROM are previable [2,3,4].

Preterm PROM after viability (24–37 weeks of gestation) is usually managed expectantly if the maternal and fetal conditions are stable [2,5]. This management has been supported by several articles, including a Cochrane review from 2017. They included 12 randomized controlled trials (3617 women) and did not find a clinically important difference in the incidence of neonatal sepsis between women who gave birth immediately and those who were managed expectantly in preterm PROM up to 37 weeks of gestation.

A recent practice bulletin from the American College of Obstetricians and Gynecologists (ACOG) also supported the expectant management of viable preterm PROM, extending the suggested gestational age for delivery from 34 to 36 weeks and 6 days in a setting of shared decision-making with the patient [2].

In contrast, the management of previable preterm PROM is not well supported by the current literature. If left to the natural course of events, up to 80% of women diagnosed with preterm PROM deliver within one week of the rupture [6]. A further prolongation of the pregnancy can expose the mother and the neonates to severe outcomes such as neonatal death and poor maternal outcomes, especially chorioamnionitis.

In 2016, a study from Ireland analyzed neonatal and maternal complications related to early preterm PROM. The average time from diagnosis to delivery was 13 days, with an average gestational age at delivery of 20 weeks and 5 days. Four fetuses died in utero, and 28 neonates died in the peripartum period. Ten infants out of forty-two survived to birth, and six of them later died in the neonatal intensive care unit (NICU) of sepsis, intraventricular hemorrhage, or respiratory distress syndrome. The overall neonatal survival to discharge was 4.76% (2/42). Maternal morbidity was low, with a rate of chorioamnionitis of 14% according to the clinical criteria and 69% according to the histological criteria [7].

In 2022, a group of researchers in Turkey analyzed 128 women with preterm PROM before 24 weeks of gestation. They found a neonatal survival rate of 60.9% and a chorioamnionitis rate of 22/128 [8].

In 2022, another article from Turkey analyzed 192 patients with a mean gestational age at previable preterm PROM of 20.45 weeks. Sixty-seven (39.2%) of them died in the neonatal period (neonatal death group), and 104 cases (60.8%) constituted the surviving neonate group. Of the surviving neonates, 37 (33.7%) experienced at least one complication. The most commonly seen maternal complications were chorioamnionitis (24.48%) and placental abruption (8.33%) [9].

In twins, the evidence is even scarcer than for singletons. A recent article from Spain retrospectively analyzed dichorionic and monochorionic diamniotic twin pregnancies complicated by preterm PROM before 24 weeks of gestation. Seven out of forty-five patients spontaneously delivered within the first 24 h after the diagnosis. Two out of forty-five patients requested the selective termination of the affected twin. In the 36 ongoing pregnancies that were managed expectantly beyond 24 h, the overall survival rate was 48.6%, and chorioamnionitis was present in 8 out of 36 patients [10].

In summary, it has been well established that previable preterm PROM neonates have lower survival rates, increased need for neonatal interventions, and higher rates of severe morbidity from complications such as sepsis, intraventricular hemorrhage, respiratory distress syndrome, and bronchopulmonary dysplasia [11,12,13]. Importantly, these complications have been shown to decrease with prolonged latency to delivery, which may motivate patients and practitioners to choose interventions to delay time to delivery as long as possible.

In addition, the placement of an amnio patch has been proposed by a few research groups. They propose the intraamniotic transfusion of maternal platelets and plasma to favor the temporary closure of the membrane defect [14,15,16]. Another alternative “experimental” solution is a serial amnioinfusion. This treatment has been the subject of a meta-analysis from 2023. This meta-analysis reviewed the effect of serial transabdominal amnioinfusion on perinatal outcomes in preterm PROM occurring before 26 weeks of gestation [17]. They could not establish clear results because of the lack of studies covering this topic.

However, these techniques are still in the experimental phase, and larger studies will be needed before widespread implementation in clinical practice can occur. Therefore, we will not focus on these procedures (amnio patch and amnioinfusion) in our article.

In contrast, poor outcomes for the mother usually increase with prolonged latency to delivery, which can place maternal and fetal well-being in an apparent juxtaposition. The maternal risks of expectant management following previable preterm PROM include but are not limited to chorioamnionitis, endometritis, placental abruption, postpartum hemorrhage, retained placenta, cesarean hysterectomy, and sepsis [7,9,12,18,19], all of which carry both short- and long-term morbidity risks.

In real-world clinical practice, given the absence of official clinical guidelines, in instances of previable preterm PROM, patients are generally offered two options: active management through the termination of the pregnancy or expectant management to prolong the pregnancy closer to fetal viability [1,2]. The counseling of patients through this difficult decision should be guided by both neonatal outcome statistics and the maternal risks associated with prolonged latency to delivery.

In the past, very little work has been carried out to directly compare maternal outcomes between active and expectant management in preterm PROM < 24 weeks of gestation. Most studies investigating previable preterm PROM only reported maternal outcomes following expectant management and used active management (elective termination) as an exclusionary factor [19,20,21,22].

Overall, there is a lack of rigorous literature review on this topic, which could very well aid in management counseling. The goal of the present meta-analysis was to determine if there was any increased risk for the mother in previous studies comparing active versus expectant management in previable preterm PROM < 24 weeks of gestation.

## 2. Materials and Methods

We searched PubMed, Google Scholar, EMBASE, and Scopus from database inception to 2 August 2022. We used the key words “preterm pre-labor rupture of membranes”, “preterm PROM”, “previable preterm pre-labor rupture of membranes”, “pPPROM”, “expectant management”, “termination”, “maternal outcomes”, “health outcomes”, and “outcomes”. We limited our search to manuscripts written in English. References from related reviews and manuscripts were searched manually to conduct a thorough review of the literature. Two investigators (G.M. and M.A.S) conducted the study selection independently, and selected studies were evaluated jointly to confirm their eligibility.

The University of Arizona institutional review board (IRB) exempted the study from approval on 30 August 2022 (STUDY00001754). This study was submitted for PROSPERO registration but was deemed ineligible as our analysis had already been completed prior to attempted registration. The study methods adhered to the PRISMA 2020 guidelines in all steps of the meta-analysis preparation [23].

The inclusion criteria for this meta-analysis were as follows: (1) the use of retrospective or prospective cohort design, (2) the inclusion of pregnant women with preterm PROM at <24 weeks of gestation, and (3) a description of maternal outcomes in active vs. expectant management groups. Studies were excluded if (1) maternal outcomes were not reported, (2) data were missing with respect to patients electing active management, (3) the data could not otherwise be extracted, or (4) no controls were reported (e.g., case reports, reviews, and surveys).

The following information and data were extracted from each qualified study: authors, location, year of publication, number of patients who chose expectant or active management, and the number of patients with poor maternal outcomes in the expectant and active management groups.

The papers included for analysis were assessed by two authors (G.M. and M.S.) for the risk of bias via the quality in prognostic score risk-of-bias assessment tool for prognostic factor studies (QUIPS) [24,25]. Each study was assessed for risks of bias in six domains: study participation, study attrition, prognostic factor measurement, outcomes measurement, study confounding, and statistical analysis/reporting. Within each domain, the studies were determined to have high, medium, or low risks of bias. If the initial determinations between the authors varied, consensus was achieved via discussion. Within each domain, if inadequate evidence was provided for appropriate assessment, a risk of bias of “low” was ruled out. The determination of a moderate versus high bias, based on the pertinence of the omission was left to the discretion of the assessors.

In addition, publication bias was represented graphically via funnel plots of the standard difference in the means versus the standard errors if there were more than three included studies regarding that specific outcome. If we could identify at least 10 articles, we were planning to perform a regression test for funnel plot asymmetry using the weighted regression with a multiplicative dispersion model. With less than 10 articles, the power is too low to distinguish change from real asymmetry via the regression test for funnel plot asymmetry. Finally, we performed the “leave one out” sensitivity analysis with the function “metainf” of the software “R” version 4.3.1 [26] for the common (fixed) effect model and random effect model.

The data analysis was performed using the software Review Manager version 5.4.1. [27] and the software “R” version 4.3.1 [26]. A meta-analysis of risk ratios with 95% confidence intervals (CIs), using the Mantel–Haenszel method, was used to explore the association between expectant management and patient health outcomes. A *p* value < 0.05 was considered statistically significant. An I-squared Higgins (I^2^) index higher than 0% was used to target potential heterogeneity. An I^2^ < 50% was considered a low level of heterogeneity. We chose to use the random effect model, considering the multiple baseline differences between the included studies.

## 3. Results

The initial key word search resulted in 22 studies, 4 of which matched the inclusion and exclusion criteria (Figure 1). Studies were largely excluded due to a lack of reported maternal outcomes following active management (elective termination). The consistently reported maternal outcome between included studies were chorioamnionitis, postpartum hemorrhage and sepsis; as such, these were the only outcomes we included for the meta-analysis.

In 2016, Wagner et al. presented a retrospective cohort analysis of women admitted to a single hospital in Germany from 2005 to 2015 [28]. This study was designed to evaluate outcomes in previable preterm PROM < 24 weeks of gestation. The exclusion criteria included multi-gestation pregnancies, documented fetal anomalies, and iatrogenic rupture of membranes following an obstetrical procedure.

They diagnosed chorioamnionitis based on maternal fever (≥38 °C), fundal tenderness, fetal tachycardia, elevated maternal white blood cell count, and C-reactive protein levels.

Chorioamnionitis was found in 30/69 (43%) patients who received expectant management, while none of the 32 patients who opted for active management had signs of chorioamnionitis at the time of termination (Table 1). A direct statistical analysis of these treatment groups was not performed by Wagner et al. [28] as they had alternative endpoints for their study.

In 2021, Simons et al. presented a prospective cohort study conducted across nine hospitals in the Netherlands [29]. This study was designed to evaluate outcomes in previable preterm PROM at <24 weeks of gestation. Patients were included if they were between 13 weeks and 24 weeks of gestation at the time of membrane rupture. Patients with iatrogenic preterm PROM following obstetrical procedures, acute infection following preterm PROM, or uterine fetal demise following preterm PROM were all included. Patients were excluded if there were documented fetal anomalies or signs of active labor prior to membrane rupture.

The authors of this study defined proved chorioamnionitis after a histopathological analysis.

Chorioamnionitis was found in 38/86 (44%) patients who elected to receive expectant management, compared to 6/12 (50%) patients who chose active management (Table 1). A direct statistical analysis of these treatment groups was not performed by the authors as they had alternative endpoints for their study.

In 2021, Pylypjuk et al. presented a retrospective cohort study conducted across two tertiary care centers in Canada using patient records from 1 January 2009 to 31 December 2015 [30]. Their study aimed to evaluate maternal and neonatal outcomes in cases of previable preterm PROM. Patients were eligible for inclusion if they underwent spontaneous preterm PROM at <24 weeks gestation. The exclusion criteria included diagnosed congenital abnormalities, iatrogenic membrane rupture, placement of rescue cerclage within 14 days of PPROM, or preterm PROM with an interval between membrane rupture and delivery of less than 24 h. They compared 74 patients who chose to undergo expectant management to 25 patients who underwent active management. They defined chorioamnionitis with placental pathological analysis after birth. Postpartum hemorrhage and sepsis were not clearly defined.

The patients who elected to receive expectant management were significantly older in age (expectant management—30.4 years; active management—25.8 years, *p* = 0.001). There was also a significant difference in gestational age at preterm PROM (expectant management—20 weeks and 6 days; active management—20 weeks and 1 day, *p* = 0.023). The authors found significantly increased risks of chorioamnionitis (*p* < 0.0001) and antepartum hemorrhage or abruption (*p* = 0.022) in the expectant management group compared to the active management group. However, no significant difference was found for other outcomes, such as postpartum hemorrhage, placenta accreta, and maternal sepsis, between groups.

In 2022, Sklar et al. presented a retrospective cohort study conducted across three university-affiliated hospitals in the U.S., using patient records from 2011 to 2018 [31]. This study was designed to evaluate a wide range of maternal health risks in previable preterm PROM < 24 weeks of gestation. Patients were eligible for inclusion if they underwent preterm PROM at 14 weeks to 23 weeks and 6 days of gestation. Pregnancies were excluded if they were complicated by chromosomal or fetal anomalies, iatrogenic rupture of membranes within 48 h of obstetrical procedure of any kind, if there were missing delivery data, or if there were specific contraindications to expectant management such as pre-existing chorioamnionitis or active heavy bleeding. They compared 108 patients who chose to undergo expectant management to 100 patients who chose to undergo active management within 48 h of membrane rupture (Table 1).

The authors defined chorioamnionitis as documented by a physician and prompting treatment with intravenous antibiotics. Postpartum hemorrhage was defined as an estimated blood loss of >1000 mL. Maternal sepsis was defined as clinical sepsis documented by a physician and evidence of infection (i.e., a fever or positive blood culture) with end-organ dysfunction (i.e., hypotension, oliguria, elevated creatinine, disseminated intravascular coagulation, decreased consciousness, or respiratory compromise).

The authors found significantly increased risks of chorioamnionitis (*p* < 0.001), postpartum hemorrhage (*p* = 0.027), and composite maternal morbidity (*p* < 0.001) in the expectant management group compared to the active management group. However, no significant difference was found for other outcomes, such as sepsis, endometritis, unplanned hysterectomy, or admission to the intensive care unit (ICU), between groups.

The QUIPS tool [24,25] revealed overall assessments of low risks of bias in Pylypjuk et al. [30] and Sklar et al. [31], a moderate risk of bias in Wagner et al. [28], and a high risk of bias in Simons et al. [29] (Figure 2). The study by Sklar et al. was deemed to have a low risk of bias in all six domains of the QUIPS tools (participation, attrition, prognostic factor measurement, outcome measurement, confounding, and statistical analysis/reporting). The study by Pylypjuk et al. was deemed to have low risk of bias in all domains except confounding bias, which was deemed to be moderate. The study by Wagner et al. was determined to have low risk of bias in 4/6 domains, and 2/6 domains (attrition and confounding bias) were scored as moderate. Simons et al. was deemed to have a low risk of bias in participation but otherwise scored with moderate/high risk of bias in all other domains. Specifically, it was determined that the statistical analysis/reporting risk of bias was high due to inconsistencies in data reporting.

Pooling together the results from the four selected studies using the random effect model, we found no statistically significant difference in the incidence of chorioamnionitis in patients electing to receive active vs. expectant management (*p* = 0.054; RR 0.30, 95% CI 0.08–1.02) (Figure 3).

A high heterogeneity of 81% (I^2^) between studies was calculated (Figure 3), and the funnel plot demonstrates asymmetry, as shown in Figure 4. The “leave one out” sensitivity analysis highlighted that without the study by Simons et al., a significant reduction in risk is seen in chorioamnionitis when active management is chosen over expectant management (Figure 5).

The number of studies available for inclusion in this analysis was too small (<10) to test for small study effects via the regression asymmetry test.

A further review of all included studies demonstrated that both the studies by Pylypjuk et al. and Sklar et al. presented data on postpartum hemorrhage and maternal sepsis; therefore, a sub-analysis was performed for these two additional maternal outcomes.

Pooling together the results of these two selected studies, no statistically significant difference was found in the incidence of postpartum hemorrhage (RR 0.75, 95% CI 0.27–2.07) (Figure 6) or maternal sepsis (RR 0.23, 95% CI 0.04–1.28) (Figure 7) in patients opting for active vs. expectant management in previable preterm PROM at < 24 weeks of gestation.. We detected a high heterogeneity of 68% (I^2^) for postpartum hemorrhage and a low heterogeneity of 0% (I^2^) for sepsis between the studies and did not create a funnel plot or carry out a regression asymmetry test because of the paucity of included studies.

## 4. Discussion

In cases of previable preterm PROM, patients are given the choice between two potential treatment paths: active management through the termination of the pregnancy or expectant management to prolong the pregnancy and increase the chance of fetal survival. During traditional counseling, the active management pathway is usually presented with fewer risks to the mother, while expectant management is thought to carry increased risks of maternal morbidity and mortality. This method of counseling is largely based on studies that have shown high risks of complications when a previable preterm PROM pregnancy is prolonged in comparison to an uncomplicated, viable pregnancy delivered at term [7,9,12,18,19]. Unfortunately, very little work has been carried out to directly compare patient health outcomes between active and expectant management, leading to limitations in a physician’s ability to provide patients with data to inform the challenging decision they face.

Interestingly, the results of the present study suggest that the risks of chorioamnionitis (Figure 3), postpartum hemorrhage (Figure 6), and maternal sepsis (Figure 7) are not significantly different between active and expectant management in previable preterm PROM < 24 weeks of gestation. This is in apparent contradiction with the previously mentioned traditional method of maternal counseling after the diagnosis of previable preterm PROM.

The lack of significance in the overall analysis of chorioamnionitis risk in active vs. expectant management is likely attributed to the high heterogeneity between studies. Specifically, the sensitivity analysis demonstrated that without the study by Simons et al., there would in fact be a significant reduction in the risk of chorioamnionitis when active management is chosen over expectant management (Figure 5), suggesting that with a larger pool of data there would likely be a significant difference seen.

The deviation in the results presented in Simons et al., when compared to the three other included studies, can be attributed to a few key study flaws. First, it is important to note that the study by Simons et al. [29] was the only included study that did not use iatrogenic rupture of membranes following an obstetrical procedure as an exclusionary factor. Iatrogenic rupture accounted for a total of 2.4% of total included patients, but data were not provided as to whether these patients elected for active or expectant management. Additionally, this study had the smallest overall sample size and the smallest group of patients who opted for active management, which decreased the overall power of the study. For these reasons and others, the study by Simons et al. was the only included study that was rated to have a high risk of bias using the QUIPS tool (Figure 2).

It is also important to consider how differences in clinical practice and cultural ideologies surrounding active management (elective termination) vs. expectant management in each country of origin (Sklar et al., United States [31]; Pylypjuk et al., Canada [30]; Wagner et al., Germany [28]; and Simons et al., The Netherlands [29]) may have impacted the patients’ decisions and physician counseling in the management of previable preterm PROM. This was aptly highlighted in in the study by Pylypjuk et al., which found a trend toward higher rates of active management at one hospital site compared to the other (31.7% vs. 15.4%, *p* = 0.070), which could have been due to differences in physician counseling between sites.

As previously mentioned, the meta-analysis presented here was limited by the scarcity of eligible studies in the current literature, with only four eligible for inclusion which all had a broad range of risk of bias, as calculated through use of the QUIPS tool (Figure 2). Additionally, the common and random effects- pooled RRs differ, consistent with the asymmetry in the funnel plot and therefore the possible selection biases within the studies.

Another limitation of the present meta-analysis was the limited number of reported maternal outcomes available in the literature. In our initial search, we found that most of the work investigating previable preterm PROM pregnancies only reported fetal outcomes. Indeed, when studies did choose to examine maternal outcomes, active management (termination) was almost always an exclusion criterion; and therefore, data on these patients’ health outcomes were not reported. The only maternal outcomes consistently reported across the four included studies were chorioamnionitis. This is despite the extensive number of potential complications of previable preterm PROM have been reported in the case report and the case series literature [7,9,12,18,19].

In fact, to our knowledge, the studies by Sklar et al. [31] and Pylypjuk et al. [30] are the only two studies in the literature which investigated further health outcomes other than chorioamnionitis (postpartum hemorrhage and maternal sepsis), directly comparing these outcomes between expectant management versus active management in previable preterm PROM < 24 weeks of gestation.

The argument could also be made that chorioamnionitis is not an appropriate health outcome to be used when evaluating risk of active management (elective termination) due to the inherent decreased risk of developing an infection during the process of labor induction or dilation and evacuation compared to a prolonged latency to delivery in expectant management. Indeed, it can be assumed that the average time to delivery is likely shorter in active management vs. expectant management and therefore further contributes to a lower risk of developing an infection.

In summary, our results are limited by the small number of available studies (four), the high heterogeneity of data, the small sample size of each study (Wagner et al.; *n* = 101 [28]; Simons et al., *n* = 98 [29]; Pylypjuk et al., *n* = 99; Sklar et al., *n* = 208 [31]), the availability of only three consistently reported health outcomes, the broad range of bias calculated by the QUIPS tool, and the bias risk visualized in the funnel plot representation of risk of bias.

Despite these limitations, this remains the first meta-analysis to directly compare patient health outcomes following active vs. expectant management in previable preterm PROM pregnancies at <24 weeks of gestation. Additionally, our study included broad search criteria to assess a wide variety of reportable patient health outcomes and adhered to the PRISMA 2020 guidelines in all steps of meta-analysis preparation [23].

## 5. Conclusions

Previable preterm PROM is a devastating diagnosis for expecting families. Unfortunately, a lack of clinical data, specifically large retrospective and prospective cohort studies, often leaves providers without clear evidence with which to counsel patients and families when deciding between active vs. expectant management following a diagnosis of previable preterm PROM.

The present study is the first of its kind to directly compare maternal outcomes in previable preterm PROM at <24 weeks of gestation via a meta-analysis. Although a statistically significant difference in the outcome of chorioamnionitis, postpartum hemorrhage, or sepsis was not detected between patients electing to receive active versus expectant management in the setting of previable preterm PROM, the analysis was limited by a scarcity of data and a large range of biases. Additionally, a sensitivity analysis highlighted that the results from one study skewed our overall results, without which there is in fact a significant reduction in the risk of chorioamnionitis in active management compared to expectant management following previable preterm PROM at <24 weeks of gestation.

While the present study should not particularly alter physician counseling in cases of previable preterm PROM, it should serve to highlight the gap in the currently available data and as a motivation for further studies comparing maternal health outcomes in active vs. expectant management to appropriately inform a patient’s decision-making process.

## Figures and Tables

**Figure 1 children-10-01347-f001:**
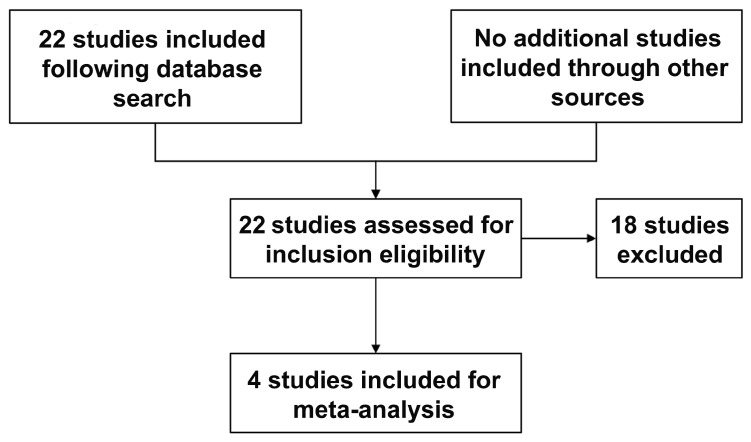
Study selection from the literature review.

**Figure 2 children-10-01347-f002:**
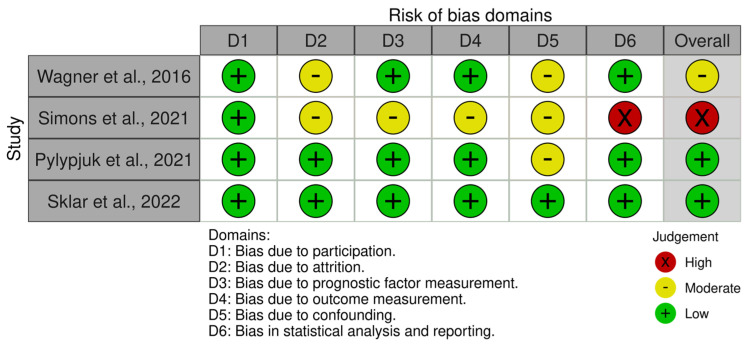
The Quality in Prognostic Studies (QUIPS) tool used to assess risk of bias. Created using the online risk of bias visualization tool “robvis” [32]. (Wagner et al., 2016 [28], Simons et al., 2021 [29], Pylypjuk et al., 2021 [30], Sklar et al., 2022 [31]).

**Figure 3 children-10-01347-f003:**
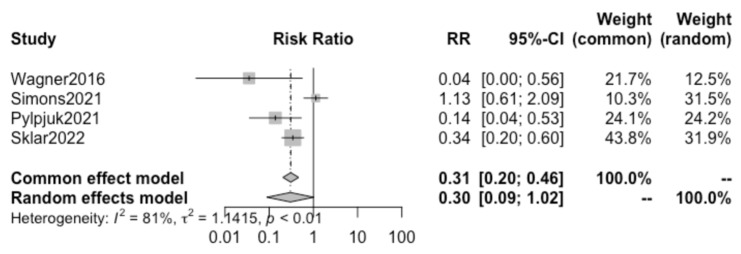
Forest plot with risk ratios for chorioamnionitis. (Wagner 2016 [28], Simons 2021 [29], Pylypjuk 2021 [30], Sklar 2022 [31]).

**Figure 4 children-10-01347-f004:**
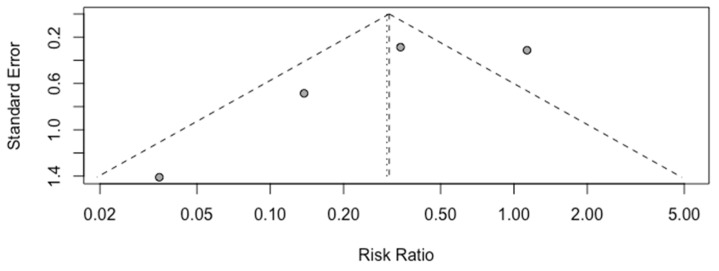
Funnel plot for risk of bias.

**Figure 5 children-10-01347-f005:**
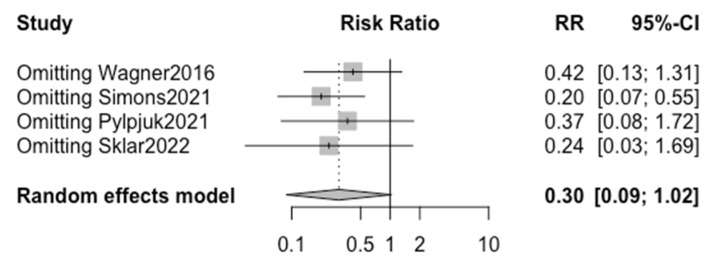
Sensitivity analysis (leave-one-out) using a random effect model for the risk of chorioamnionitis. (Wagner 2016 [28], Simons 2021 [29], Pylypjuk 2021 [30], Sklar 2022 [31]).

**Figure 6 children-10-01347-f006:**
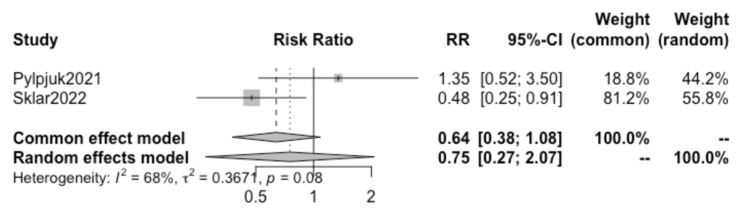
Forest plot with PPH risk ratios (Pylypjuk 2021 [30], Sklar 2022 [31]).

**Figure 7 children-10-01347-f007:**
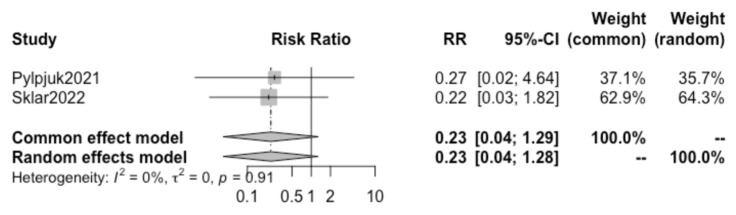
Forest plot with risk ratios of sepsis (Pylypjuk 2021 [30], Sklar 2022 [31]).

**Table 1 children-10-01347-t001:** Demographic information and maternal outcomes from included studies. “*n*” = number. “n/a” = no data available for analysis.

Authors, Year	Location	Study Design	Subjects (Total *n*)	Expectant *n* (%)	Active *n* (%)	Chorioamnionitis in Expectant *n* (%)	Chorioamnionitis in Active *n* (%)	Postpartum Hemorrhage in Expectant *n* (%)	Postpartum Hemorrhage in Active *n* (%)	Sepsis in Expectant *n* (%)	Sepsis in Active *n* (%)
Wagner et al., 2016 [28]	Germany	RCS	101	69 (68%)	32 (32%)	30 (43%)	0 (0%)	n/a	n/a	n/a	n/a
Simons et al., 2021 [29]	The Netherlands	PCS	98	86 (87%)	12 (13%)	38 (44%)	6 (50%)	n/a	n/a	n/a	n/a
Pylypjuk et al., 2021 [30]	Canada	RCS	99	74 (75%)	25 (25%)	43 (58%)	2 (8%)	11 (15%)	5 (20%)	5 (7%)	0 (0%)
Sklar et al., 2022 [31]	United States	RCS	208	108 (51%)	100 (49%)	41 (38%)	13 (13%)	25 (23.1%)	11 (11%)	5 (4.6%)	1 (1%)

## Data Availability

No new data were created or analyzed in this study. Data sharing is not applicable to this article.

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
