# Peer review of "Maternal Outcomes Following Active vs. Expectant Management of Previable Preterm Pre-Labor Rupture of Membranes: A Meta-Analysis"

_children, 2023, doi:10.3390/children10081347_

Round 1

Reviewer 1 Report

Dear authors,

In this review, you wanted to determine whether there was any difference in maternal risks when active versus elective management of preterm pre-labour rupture of membranes was chosen.

The review will be of importance to the relevant community. The submitted paper is generally well-composed.

Introduction: The introduction is well organized and presents relevant information for the content of the article.

Materials and Methods: are well structured, with an adequate description of study design, searches were made in PubMed, Google Scholar, EMBASE and Scopus.

Furthermore, the results are presented clearly with pertinent table and are discussed in relation to recent literature.

Although the results are limited by the small number of studies (3), the high heterogeneity of the studies (I2 = 83%), the small sample size of each study, the availability of only one health outcome, chorioamnionitis, the conclusions are drawn from the systematic review's results and bear clinical relevance.

Author Response

Dear Reviewer,

We appreciate your review and we would be honored to continue the submission process.

Best regards,

The authors

Reviewer 2 Report

Thank you for submitting your manuscript entitled “Maternal outcomes active vs. expectant management of pre-viable preterm pre-labor rupture of membranes: a meta-analysis ”.  I have carefully reviewed the manuscript. I want to provide you with feedback and recommendations.

Please find below my comments:

·      It is worth considering how and when this manuscript will be of interest and provide valuable information to the readers of “children” given that the title of the specific journal refers to a different age group rather than pregnant women. 

·       It is advisable to reassess the study design to ensure its validity.   

·       To enhance the quality of the review, the authors have to provide further elucidation on the definition of chorioamnionitis and the other clinical entities utilized in the included articles.     

Minor editing of the English language required

Author Response

Dear Reviewer,

Thank you for taking the time to review our manuscript. Please find below our reponse.

- It is worth considering how and when this manuscript will be of interest and provide valuable information to the readers of “children” given that the title of the specific journal refers to a different age group rather than pregnant women.

- The article would fit the scope of this special issue titled "Amniotic Fluid Anomalies: From Prenatal Management to Neonatal Outcomes" and it has been invited by the Editors of the journal.

- It is advisable to reassess the study design to ensure its validity.

- We reassessed the study design and we established its validity.

- To enhance the quality of the review, the authors have to provide further elucidation on the definition of chorioamnionitis and the other clinical entities utilized in the included articles.   

- We thank you for your feedback on our manuscript and have added a description of the clinical entities (outcomes) as defined by the single studies. Please see the edited result section of the manuscript.

We hope we addressed all of your comments.

Best regards,

The Authors.

Reviewer 3 Report

This is an extremely useful study for clinicians worldwide, when dealing with previable PPROM. It is unfortunate that more study on this topic could not be found and also, the limited number of reported outcomes is disappointing. This, however, does not impact the quality of the manuscript, which is well written and it has clear objectives and analysis.

There is one major point I wish to make: I find that, as long as we are talking about ”maternal” outcomes (as it should be) and as long as we use the term pregnant ”women” on row 83 (as it should be), the paragraph in rows 53-57 has absolutely NO place in this manuscript. This manuscript does not deal with social issues regarding equal rights for the LGBTQIA+ individuals, but attempts an objective analysis of three studies in which no such individual seem to be involved – in all three studies, the patients/subjects are described as ”women”.

Minor issues:

- the closing parenthesis on row 188 should be placed earlier in the sentence – ”(attrition and confounding bias)”

- I don’t think I saw Figure 2 referred anywhere in the manuscript. Also, in the final version of the article, the figures should be better balanced in terms of text size – in figures 1 and 4 the text is overwhelmingly large.

Author Response

Dear Reviewer,

Thank you for taking the time to review our manuscript.

Please find below our response.

- This is an extremely useful study for clinicians worldwide, when dealing with previable PPROM. It is unfortunate that more study on this topic could not be found and also, the limited number of reported outcomes is disappointing. This, however, does not impact the quality of the manuscript, which is well written and it has clear objectives and analysis.

- We thank you for your feedback on our manuscript, and hope that it will in fact bring light to how lacking the current literature is on this subject.

- There is one major point I wish to make: I find that, as long as we are talking about ”maternal” outcomes (as it should be) and as long as we use the term pregnant ”women” on row 83 (as it should be), the paragraph in rows 53-57 has absolutely NO place in this manuscript. This manuscript does not deal with social issues regarding equal rights for the LGBTQIA+ individuals, but attempts an objective analysis of three studies in which no such individual seem to be involved – in all three studies, the patients/subjects are described as ”women”.

- We welcome this feedback and have happily removed the paragraph from the introduction of our manuscript.

Minor issues:

- the closing parenthesis on row 188 should be placed earlier in the sentence – ”(attrition and confounding bias)”

- Thank you for highlighting this error, we appreciate your thorough review. We edited the text accordingly.

- I don’t think I saw Figure 2 referred anywhere in the manuscript.

- Please see line 155, 259, 324, 337.

- Also, in the final version of the article, the figures should be better balanced in terms of text size – in figures 1 and 4 the text is overwhelmingly large.

- Thank you for this suggestion. We will work with the editing staff to ensure that all figures have appropriate formatting prior to publication.

We hope we responded to all your comments.

Best regards,

The Authors.

Reviewer 4 Report

The authors conducted a meta-analysis of maternal outcomes following active versus expectant management of PPROM. The topic is current, the manuscript is well-written and organized. As the authors stated, the small number of studies and the high heterogeneity of the studies are major limitations of the manuscript, but the importance of the topic exceeds these limitations. I recommend the acceptance in the present form. 

Author Response

Thank you for your comments,

Respectfully,

The Authors

Round 2

Reviewer 1 Report

Dear authors,

In this review, you wanted to determine whether there was any difference in maternal risks when active versus elective management of preterm pre-labour rupture of membranes was chosen.

The review will be of importance to the relevant community. The submitted paper is generally well-composed.

Introduction: The introduction is well organized and presents relevant information for the content of the article.

Materials and Methods: are well structured, with an adequate description of study design, searches were made in PubMed, Google Scholar, EMBASE and Scopus.

Furthermore, the results are presented clearly with pertinent table and are discussed in relation to recent literature.

Finally, I recommend accepting this review in present form.

Reviewer 2 Report

The authors in their manuscript entitled “Maternal outcomes active vs. expectant management of pre-viable preterm pre-labor rupture of membranes: a meta-analysis” aimed to determine whether there was any difference in the risk related to the mother's health when active versus elective management of preterm pre-labor rupture of membranes. 

Despite the significant limitations of the small number of studies and high heterogeneity, the importance of the topic outweighs these limitations. The manuscript is well written. Therefore, I recommend the acceptance in the present form. 

Reviewer 3 Report

It is salutary that the authors took their time to include a whole different study to their analysis. This makes their research even more impactful. Even though the Introduction section is a bit long and deals with issues not related to the subject of the manuscript (viable PPROM), I agree that a comparison is maybe necessary. I am convinced that specialists will benefit from reading this article and I agree with its publication as it is.

Reviewer 4 Report

As I already stated in the first round of review, the manuscript should be considered for the publication in the present form